# Coronary Artery Ectasia: Review of the Non-Atherosclerotic Molecular and Pathophysiologic Concepts

**DOI:** 10.3390/ijms23095195

**Published:** 2022-05-06

**Authors:** Gavin H. C. Richards, Kathryn L. Hong, Michael Y. Henein, Colm Hanratty, Usama Boles

**Affiliations:** 1Cardiovascular Research Institute (CVRI) Dublin, Mater Private Hospital, D07 WKW8 Dublin, Ireland; gavin.richards@materprivate.ie (G.H.C.R.); kathryn.hong@ucdconnect.ie (K.L.H.); colm.hanratty@materprivate.ie (C.H.); 2School of Medicine, University College Dublin, D04 V1W8 Dublin, Ireland; 3Department of Public Health and Clinical Medicine, Heart Clinic, Umea University, 901 87 Umea, Sweden; michael.henein@umu.se; 4Cardiology Department, Tipperary University Hospital, E91 VY40 Clonmel, Ireland

**Keywords:** coronary artery ectasia, coronary artery aneurysm, CAD, cytokine, lipidome

## Abstract

Coronary artery ectasia (CAE) is frequently encountered in clinical practice, conjointly with atherosclerotic CAD (CAD). Given the overlapping cardiovascular risk factors for patients with concomitant CAE and atherosclerotic CAD, a common underlying pathophysiology is often postulated. However, coronary artery ectasia may arise independently, as isolated (pure) CAE, thereby raising suspicions of an alternative mechanism. Herein, we review the existing evidence for the pathophysiology of CAE in order to help direct management strategies towards enhanced detection and treatment.

## 1. Introduction

Coronary artery ectasia (CAE) is a relatively common coronary angiographic finding, with an incidence of 1.5–5% and geographical variations in prevalence. CAE has been associated with a male predominance (1.7% vs. 0.2%) and more frequently affects the right coronary artery and the proximal vessels. Currently, the pathogenesis of CAE is not fully understood, with some evidence suggesting an atherosclerotic aetiology and other reports describing a distinct pathology [1].

While the terms CAE and coronary artery aneurysm (CAA) are often used interchangeably, they carry distinct phenotypes and definitions. CAE is defined as a diffuse dilatation of the coronary artery of at least 1.5 times the normal artery with a length of over 20 mm or greater than one third of the vessel. It can be further subdivided into diffuse and focal dilations by the length of the dilated vessels. Histologically, it presents with extensive destruction of musculoelastic elements, with marked degradation of collagen and elastic fibres and disruption of the elastic lamina. Conversely, CAA is a dilatation with a focal appearance. It is termed saccular if the transverse diameter is greater than the longitudinal, and fusiform for the opposite [2].

Given that CAE is associated with atherosclerosis in 50% of cases, a common underlying aetiology has been postulated. It is important to note, however, that approximately 30% of cases are associated with vasculitis including Kawasaki disease and Takayasu arteritis, and connective tissue diseases such as Ehlers–Danlos or Marfan’s syndrome. The remainder are congenital or idiopathic [3]. In particular, congenital CAE has been documented with other cardiovascular abnormalities including bicuspid aortic valve, aortic aneurysms, pulmonary stenosis, and ventricular septal defects.

## 2. Acute Myocardial Infarction in CAE

CAE patients with acute myocardial infarction (MI) undergoing percutaneous coronary intervention (PCI) have reported a high thrombus burden and greater use of glycoprotein IIb/IIIa inhibitors (GPI) and post-procedural anticoagulation [4]. Indeed, thrombus formation may be inherently related to abnormal flow within coronary ectatic lesions, resulting in distal embolization. While higher rates of no-reflow and lower Thrombolysis in Myocardial Infarction (TIMI) flow grades have been observed after percutaneous coronary intervention (PCI), long term survival is good [5]. One study reported that no stent was deployed in 44% of patients undergoing coronary angiogram, compared with 7.5% in a comparable group with no CAE [6]. Furthermore, stent deployment conferred a better in-hospital outcome, although long term outcomes were shown to be similar with relatively high rates of non-fatal MI and angina. In a large observational study evaluating the long-term outcomes of 1698 patients with acute MI, the incidence of major adverse cardiac events (MACE) (*p* < 0.001), cardiac death (*p* = 0.004), and non-fatal myocardial infection (*p* < 0.001) was significantly higher in patients with CAE [7]. An increased risk of MACE has also been identified in patients with diffuse CAE as compared to those with focal CAE [8].

## 3. Clinical Sequelae

While most patients with CAE experience coronary artery disease (CAD), additional clinical manifestations may be related to increased inflammatory markers in the peripheral blood [9] and anomalies present in other blood vessels [10]. Possible aetiologies contributing to the destruction of musculoelastic coronary elements in CAE include vascular endothelial dysfunction, oxidative stress [11], and enzyme destruction [12].

Angina is a frequently reported symptom, resulting from slow coronary flow due to turbulence within ectatic segments. Disturbed blood flow patterns directly affect endothelial cells by promoting the sustained activation of atherogenic genes, such as monocyte chemotactic protein-1 (MCP-1). This subsequently induces monocyte infiltration and platelet-derived growth factors, further increasing endothelial cell turnover and smooth muscle cell migration [13]. Previous reports have even elucidated a higher prevalence in CAE compared with matched patients with severe atherosclerotic CAD [14]. Taken together, patients with CAE may have a higher relative risk for angina and associated risk of adverse cardiac-related outcomes when coupled with obstructive CAD.

The incidence of PCI in CAE with CAD is significantly lower than CAD with no CAE (*p* < 0.001), reflecting the limitations of current technologies to treat ectatic vessels. Coronary embolization in CAE can result from stasis in dilated segments and anticoagulation is frequently prescribed to mitigate this. Rupture of aneurysmal segments is a rare but serious complication [2].

CAE is also associated with ECG markers of arrhythmia including QRS fragmentation [15]. Conlon et al. reported an association between CAE and ECG markers of arrhythmia including prolonged Tp-Te, QTc dispersion, and P wave dispersion. Long Tp-Te interval represents a susceptibility to ventricular arrhythmias and is associated with increased mortality in hypertrophic cardiomyopathy, myocardial infarction, and long QT syndromes [16].

Long term follow-up studies have demonstrated higher incidence of acute coronary syndrome (ACS) in CAE compared to controls, with increasing ACS in higher grade CAE (Markis grades 1 and 2) [17]. Mortality and cardiovascular mortality are also higher in CAE when compared with controls [18], and there is evidence for the role of dilatation extent in predicting these clinical outcomes.

## 4. Risk Factors

A recent study demonstrated no difference in the incidence of hypertension, diabetes, hyperlipidaemia, family history, or smoking between CAE and CAD. However, higher incidences of hypertension, hyperlipidaemia, triglyceride, and low-density lipoprotein/high-density lipoprotein ratio (LDL/HDL) have been observed in previous studies when CAE patients were compared with matched controls (*p* < 0.001) [19]. This is in accordance with recent evidence describing reduced HDL-C and higher TG/HDL-C monocyte/HDL-C ratios in CAE and CAD groups as compared to controls. The LDL-C/HDL-C ratio was also significantly higher in patients with CAE versus those with CAD [20]. Furthermore, patients with solely (pure) CAE have been found to be younger, have diffuse disease involving the three main epicardial coronary branches, and have less traditional CV risk factors than those with mixed CAE [21].

While coronary angiography is the gold standard diagnostic technique for detecting CAE, intravascular ultrasound is frequently used to confirm CAE morphology and luminal dilatation. To further classify anatomical variations, Markis proposed a classification of CAE based on the extent of ectactic involvement. As described in Table 1, severity type decreases from Type I, diffuse ectasia of two or three vessels, to Type IV, localized or segmental ectasia.

## 5. Atherosclerotic vs. Non-Atherosclerotic Inflammatory Response in CAE

Given similar histological characteristics, clinical symptoms, and disease co-existence, an atherosclerotic process has been widely linked to CAE pathogenesis. Indeed, CAE may represent an exaggerated form of extensive vascular remodelling in response to atherosclerotic plaque formation, with extracellular enzymatic degradation playing a major role in ectatic vessel formation. The atherosclerotic process may extend through the intima to the media where hyalinisation and lipid deposition in the intima leads to degradation of the media due to overexpression of matrix metalloproteinases (MMPs). Consequently, MMPs are actively involved in the proteolysis of extracellular matrix proteins, resulting in collagen degradation and pathological dilatation. The overproduction of MMPs may lend itself to the development of ACS and may explain the beneficial role of rosuvastatin in suppressing MMP expression and reducing inflammation in CAE patients [22]. Of note, MMP expression is downregulated in diabetes, which may paradoxically explain the lower incidence of CAE in diabetes.

Local coronary flow disturbances caused by decreased endothelial shear stress has also been proposed as an alternative explanation for the coexistence of CAD and CAE. Intravascular ultrasound (IVUS) evidence suggests that atherosclerotic plaques within ectatic regions of vessels are highly inflamed and meet high-risk plaque criteria [23]. Histopathological evidence of CAE shows intense proteolysis and extracellular matrix destruction within the vascular wall [14].

On the other hand, risk factors for CAD are not a prerequisite for the development of CAE, and many patients are found to have no atherosclerotic plaque. Namely, Kawasaki disease is the second most common aetiology in CAE, presenting with diffuse infiltration of the arterial wall by mononuclear cells, lymphocytes, and macrophages [24]. Moreover, infection-linked CAE is associated with pathogenic invasion of the coronary arteries and immune complex deposition.

## 6. Immuno-Inflammatory Response in CAE

Mediators of chronic inflammation, such as growth factors and cellular adhesion models, have been widely described in the pathogenesis of CAE. Specifically, the expression of specific inflammatory markers, particularly IL-6 and CRP, is known to be higher in CAE compared with CAD and healthy controls [9]. Most recently, a large meta-analysis elucidated the role of other contributory markers, neutrophil to lymphocyte ratio (NLR) and red cell distribution width (RDW), in the pathogenesis of CAE [25].

A report on immune-inflammatory response in CAE demonstrated significantly higher systemic levels of INF-gamma, TNF-alpha, IL-1ß, and IL-8, and lower levels of IL-2 and IL-4 compared with the control group [26]. In comparison with CAD, CAE patients had significantly higher levels of IL-8 and IL-1ß, and significantly lower levels of IL-2 and IL-4. Analysis of isolated CAE versus mixed CAE did not demonstrate any differences with respect to cytokine levels.

Inflammatory markers, C-reactive protein and albumin are believed to be involved in the progression and severity of CAE. Recently, a significantly higher C-reactive protein-to-albumin ratio has been associated with isolated CAE when compared to obstructive CAD and controls. Notably, C-reactive protein-to-albumin ratio also correlated strongly with the severity of CAE, which provides further evidence for its potential role in detection and management [27].

While the cytokine milieu in CAE shares similarities with CAD, there are some distinct differences. Notably, the higher presence of leukocytes in CAE presents as higher levels of IL-6 and lower levels of IL-2. This cytokine-mediated inflammatory response serves as the basis for impaired coronary circulation.

High levels of TNF-α are known to be present in CAD where stimulation of the Th1 pathway leads to activated M1 macrophages which promote atherogenesis. CAE in the absence of CAD is also associated with high levels of TNF-α, which may imply a common mechanism of macrophage activation. Nevertheless, the low level of IL-2 in CAE may suggest an alternative trigger for the direct activation of the Th1 pathway.

The pro-inflammatory marker, IL-6, has a role in inhibiting macrophage activation by inhibiting macrophage scavenger receptor-A, but is not associated with CAD [28]. Additionally, the non-atherogenic process in CAE may be partially explained by lower levels of IL-4, secreted by Th2. Likewise, the lower levels of IL-2 seen in CAE may also support a non-atherogenic pathway through the absence of Th1 cell response which is associated with CAD and acute coronary syndrome.

Another report [29] found some subtle differences in cytokine levels. Triantafyllis et al. reported high IL-4 and low IL-2 levels in CAE compared with CAD and control subjects, and high IL-6 in CAE and CAD compared with control subjects. They concluded that Th2 activation (in the presence of high IL-4) is a cardinal feature of CAE (Figure 1, Pathway A). The relationship of Th2 with atherogenesis is complex as IL-4 produced by Th2 reduces IFN-y activity and so can be considered antiatherogenic. However, in some circumstances in mouse models, IL-4 was found to be associated with the promotion of atherosclerosis. Other Th2 related cytokines such as IL5 and IL33 are antiatherogenic.

In contrast, Bose et al. reported low IL-4 and IL-2 and high IL-6 in CAE compared with CAD and control groups. They proposed that the activation of smooth muscle cells by IL-6 leads to vascular remodelling and, in the absence of M2 macrophages to limit tissue damage, leads to the development of CAE (Figure 1. Pathway B) [30].

A similar histological examination of CAE and CAD supports the notion that ectasia may be a variant of atherosclerosis. Histology demonstrated extensive destruction of the musculo-elastic element of the vessel wall with degradation of medial collagen and disruption of the internal and external elastic lamina. The expansive remodelling of the external elastic membrane, likely due to the activation of MI macrophages, underlies luminal expansion in CAE. Specifically, elevated TNFa and IFNy from activated M1 macrophages promote macrophage transmigration via ICAM1 and VCAM1 into the intima and induce MMPs that inhibit collagen synthesis.

Further evidence for a common underlying pathogenesis with atherosclerosis emerges from recent reports demonstrating MCP-1 as an independent predictor of CAE. As previously mentioned, MCP-1 is directly responsible for disturbed blood flow and is critical in the development of atherosclerosis, specifically with regard to the recruitment of monocytes into the vascular wall. Additionally, higher levels of MCP-1 have also been associated with higher incidence of acute ischemic events in patients with CAD [13].

Of note, significantly elevated mean platelet volume (MPV) has also been observed with CAE and CAD groups compared to controls. Elevated MPV levels result from increased platelet activation and therefore predispose patients to a higher risk of thrombotic events and myocardial infarction [20].

## 7. Lipid Profiling in CAE

Lipoproteins have been implicated in the remodelling process leading to the development of CAE; however, evidence remains highly elusive. Lipids are known to play an important role in the formation of CAD in the presence of inflammation and oxidative stress. In addition to well-known lipid classes associated with CAD, such as cholesterol and triglycerides, advancements in lipidomic profiling have demonstrated additional lipid classes that are strongly associated with CAD, such as phospholipids [31]. Distinct patterns of individual lipid species within the phospholipid class can potentially differentiate stable from unstable CAD [32].

Previously, a higher prevalence of CAE has been demonstrated in patients with familial hypercholesterolemia alongside higher LDL-C levels, lower HDL-C, and a higher LDL/HDL ratio [19]. Likewise, an elevated LDL-C/HDL-C ratio carries predictive value for CAE development [33]. LDL-C binds to elastin, collagen, and proteoglycans and is subsequently oxidized, hence increasing its affinity to matrix particles. Foam cell formation, and the subsequent extracellular matrix (ECM) breakdown results from the stimulation of macrophages, smooth muscle cells, and ECM-degrading enzymes including MMP-2, MMP-9, and MMP-12 [23]. This is critical in the pathogenesis of dilating vascular diseases, such as CAE, as the ECM provides structural support to the vessel wall, thereby influencing cell behaviour and signalling.

By a similar mechanism, two phospholipid species, specifically sphingomyelin (SM) and phosphatidylcholine (PC), have been found to be significantly downregulated in CAE compared with CAD and healthy controls. In the first lipid profiling study reported for CAE, lipidomic profile in CAE demonstrated distinct patterns of lipid species compared with CAD and healthy controls [34]. SM are carried into the vessel wall on lipoproteins and stimulate foam cell formation [30]. They have also been shown to be incorporated in atherosclerotic plaques in addition to the polyunsaturated cholesteryl esters of long-chain fatty acids. SM levels are independently predictive of the presence of atherosclerotic CAD, a finding that implicates SM in the process of atherosclerotic plaque formation. Notably, low levels of SM in CAE may predispose premature apoptosis within the arterial wall, further promoting ectasia. Lower phosphatidylcholine levels, specifically 16-carbon fatty acyl chain phosphatidylcholines, have also been implicated in the pathogenesis of CAE. PC has a critical role in transporting fatty acids as well as lipid metabolism; thereby providing important knowledge about lipid regulation and disease manifestation. Taken together, downregulated phosphatidylcholine levels and a distinct lipidomic profile in CAE may suggest a non-atherogenic origin of ectasia development.

Similarly, subsequent work in metabolic characterization of fatty acids in patients with CAE can be distinguished from those of controls and CAD. This provides further lipidomic profiling findings that isolated plasma fatty acids profiles with CAE could be seen as biomarkers to distinguish CAE from controls and CAD patients [35].

On the grounds of the aforementioned studies, it may be adequate to conclude that the CAE lipid profile has a clearly distinct pathophysiology from atherosclerosis profile in CAD and hence a distinguished metabolic pathway. The proposed mechanistic pathways underlying the pathogenesis of CAE are depicted in Figure 2.

## 8. Treatment Options of CAE

Coronary ectasia management is highly dependent on clinical experience and the diagnosis of the disease remains largely incidental, especially for isolated forms of the disease without a preceding history from childhood or adolescence. Adding to this challenge, consensus treatment guidelines have yet to be established, despite increasing evidence of long-term outcomes related to CAE. Available management options include pharmacologic therapy, surgery, and percutaneous intervention.

Given that high grade CAE may predispose to thromboembolic-related ACS, formal anticoagulation has been proposed as a potential treatment strategy. Nevertheless, there still exists a lack of quality data to support this recommendation, and the use of anticoagulation remains controversial.

In the presence of CAD, the prognosis and treatment of CAE are similar to CAD alone. In isolated CAE, however, prognosis is better and antiplatelets are the mainstay treatment option [36]. A rationale for antiplatelet therapy in the absence of atherosclerotic coronary disease may be that platelet activation in isolated CAE is heightened through P-selectin, beta-thromboglobulin and platelet factor 4. Furthermore, given that patients with CAE have been found to have significant elevations of MPV compared to their healthy counterparts, anti-thrombotic therapy may have a beneficial role in management. In general, anticoagulation is recommended with extensive CAE and multi-vessel involvement. In the context of CAE with acute coronary syndrome with obstructive atherosclerotic CAD, the management options should follow the standard guidelines for revascularization.

Nitrates may promote dilatation and potentially exacerbate turbulent flow with a theoretical worsening of ischaemic symptoms, so the use of nitrates is generally avoided.

As statins can inhibit the secretion of MMPs, they may have a therapeutic role in the management of CAE, especially in younger patients. Higher inflammatory marker levels, specifically IL-6 and CRP, have also been observed in younger CAE patients, who have subsequently benefited from rosuvastatin therapy [22].

Overall, standard treatment of co-existent CAD is recommended, including lifestyle modification strategies and cardiovascular risk factor management. For patients with coexisting obstructive lesions and symptoms of significant ischemia despite medical intervention, percutaneous or surgical vascularization may be recommended [37]. Previous reports have described the use of coronary artery bypass grafting for the treatment of significant CAD co-existing with ectatic coronary segments [38].

Most recently, evidence for the prognostic role of serum DAMPs in the pathogenesis of CAE has emerged. The differential regulation of DAMPs S100B, HSP70, DJ-1, and sRAGE in CAE adds to the growing body of literature for novel biomarkers as therapeutic targets in CAE management [39].

## 9. Epigenetics in CAE Pathogenesis

Emerging evidence has focused on the role of epigenetics and gene regulation in the pathophysiology of cardiovascular disease (CVD) and aging. With regard to CAE, epigenetic modifications have been observed with endothelial cells, vascular smooth muscle cells, and macrophages, in addition to the inflammatory processes governing CAE development [40].

To date, the epigenetic contribution to CAD involves histone modifications, DNA methylation, and RNA-based mechanisms. Interestingly, many of the pro-inflammatory genes involved in CAE pathogenesis, IFN-y and IL-6, appear to be regulated through DNA methylation [41]. In a study investigating the relationship between IL-6 methylation status and CVD risk, lower methylation levels were found in patients with CAD, suggesting an inverse relationship between methylation and CAD risk [42]. Given the tissue heterogeneity of atherosclerosis, the precise mechanisms of DNA methylation in atherosclerosis remain to be elucidated.

Due to the impact of foam cell formation in the pathogenesis of CAE, it is important to note that several cellular microribonucleic acids (miRNAs) have been identified in this process. miRNAs play a role in the inhibition of macrophage cholesterol efflux via ABCA1 [43] as well as regulating the balance between pro-atherosclerotic M1 and anti-atherosclerotic M2 phenotypes specifically via miR-33 [44]. For these reasons, future research should explore circulating miRNAs as potential diagnostic biomarkers in various CAD settings.

As the role of genome sequencing and metabolomics continues to evolve, there is also an increasing need to explore gut microbiome dysbiosis in the progression of CAD. Multiple studies have identified microbial stains associated with CAD [45,46], while alterations in the gut microbiome have been related to the development of several cardiovascular risk factors including diabetes mellitus [47] and obesity [48]. Interestingly, a decrease of *Bacterioidetes* and increase of *Firmicutes*, *Escherichia-Shigella*, and *Enterococcus* in the gut of CAD patients suggests that a shift in microbiota may underpin the development of CVD [49].

## 10. Gut Microbial Metabolites in CAE

Metabolites of the gut microbiome, such as trimethylamine (TMA), may promote atheromatous plaque formation through metabolic processes in the systemic circulation. TMA is generated by the gut microbiota from dietary phosphatidylcholine and carnitine and oxidized in the liver to form TMAO, which results in forma cell aggregation [50]. Elevated serum levels of TMAO have been associated with early atherosclerosis, increased risk of CVD mortality, and severity of peripheral artery disease [51]. A larger-scale prospective study established the prognostic value of TMAO in the identification of patients at risk for incident adverse cardiac events at 5 years, including MI and MACE (MI and cardiovascular death) [52].

However, gut microbiome dysregulation decreases the expression of tight junction proteins, thereby increasing the permeability of the intestinal mucosa and allowing Gram-negative bacterial lipopolysaccharide (LPS), or endotoxins, to enter the blood circulation [50]. This triggers the expression of chemokines and cell adhesion molecules, further stimulating the formation of foam cells and the adhesion of monocytes to endothelial cells. Indirectly, bacterial LPS triggers the release of many pro-inflammatory factors involved in CAE pathogenesis, such as TNF-a and IL-1B, while inhibiting the expression of cholesterol transporters [53].

This explanation supports the different metabolite profiling for CAE from both atherosclerotic and normal coronaries [34].

## 11. Conclusions

While CAE remains an important clinical coronary pathology with associated morbidity and mortality, its exact underlying etiopathogenesis has yet to be fully elucidated. Due to its strong association with coronary atherosclerosis, a heightened immuno-inflammatory response is largely believed to contribute to its pathogenesis. However, CAE can develop in the presence or absence of CAD, suggesting that there may be more than one mechanism involved. Indeed, there are significant differences in the cytokine milieu and the lipidomic profile in CAE compared with CAD and healthy controls, which strongly suggests a distinct pathogenesis is at play, in addition to the development of CAD. Recent evidence has strengthened postulates of an inflammatory mechanism by establishing the role of novel inflammatory markers such as MCP-1 in the pathogenesis of CAE. The mainstay treatment of CAE includes optimal management of coexistent CAD and antiplatelet therapy for the treatment of isolated CAE. Earlier studies confirmed the role of anticoagulant in extensive CAE, however, further studies are warranted to confirm that at larger scale of patients. Novel biomarker targeted therapy in CAE management is considered the ultimate effective approach. CAE has a proven different aetiopathology, metabolites profile, worse clinical prognosis and anatomical appearance than simply atherosclerotic corornaries and hence CAE may represent a different disease.

## Figures and Tables

**Figure 1 ijms-23-05195-f001:**
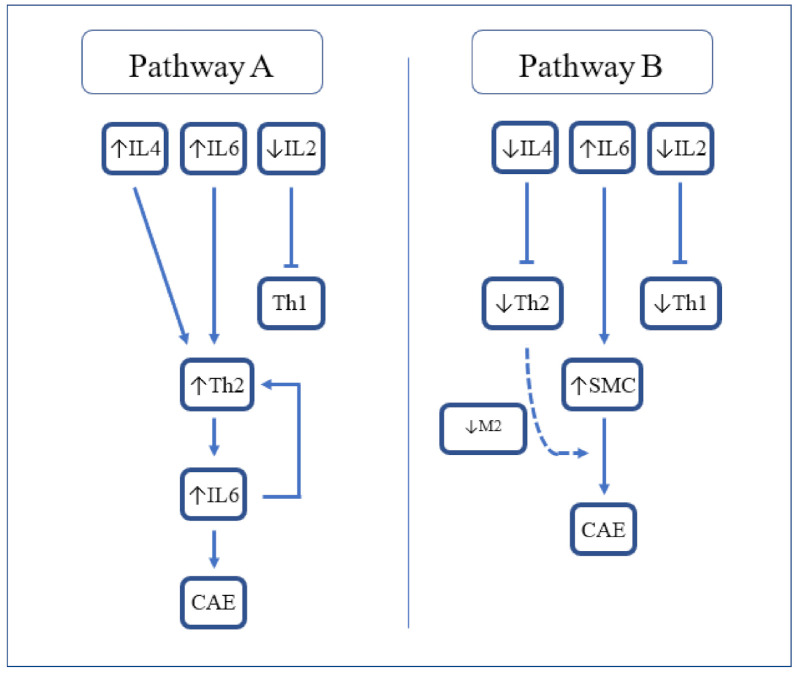
Proposed cytokine-mediated pathways resulting in CAE. IL-2 = Interleukin-2; IL-4 = Interleukin-4; IL-6 = Interleukin-6; Th2 = T-helper 2; CAE = coronary artery ectasia; SMC = smooth muscle cells; and M2 = M2 macrophages.

**Figure 2 ijms-23-05195-f002:**
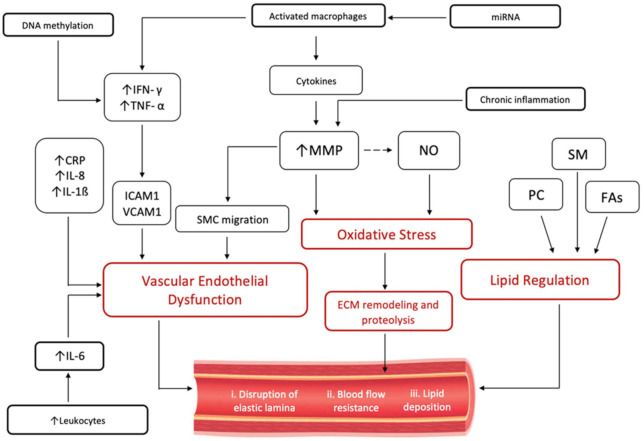
Schematic of proposed pathways underlying vascular remodelling in the pathogenesis of CAE. Proposed triggers such as e.g., viral or Gut microbial metabolites are not included in the figure. IFN-γ = interferon gamma; IFN-α = interferon alfa; IL-6 = interleukin-6; IL-8 = interleukin-8; IL-1β = interleukin 1-beta; CRP = c-reactive protein; ICAM1 = Intercellular Adhesion Molecule 1; VCAM1 = Vascular Cell Adhesion Molecule 1; MMP = Matrix Metalloproteinases; SMC = smooth muscle cells; NO = nitric oxide; ECM = extracellular matrix; SM = sphingomyelin; PC = phosphatidylcholine; FAs = fatty acids; and miRNA = microribonucleic acids.

**Table 1 ijms-23-05195-t001:** Markis classification of coronary artery ectasia.

Type I	Diffuse ectasia of two or three vessels
Type II	Diffuse disease in one vessel only and localised in another vessel
Type III	Diffuse disease in one vessel
Type IV	Localised or segmental ectasia

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
