# Peer review of "Coronary Artery Ectasia: Review of the Non-Atherosclerotic Molecular and Pathophysiologic Concepts"

_ijms, 2022, doi:10.3390/ijms23095195_

Round 1
Reviewer 1 Report
I think that the manuscript is very interesting and extensively reviews the possible mechanisms implicated in CAE that are common to CAD and also proposes alternative mechanisms that could help to distinguish between both diseases and to establish a specific treatment. For all these reasons, I think that the review is publishable in the special issue.
Author Response
We thank the author for taking the time to review our manuscript and providing positive feedback.
Reviewer 2 Report
1. A major concern are missing references throughout the manuscript. Several important statements by the authors are not clearly supported by underlying citations or identified as comments: f.exp.
-
-
p.6: “Overall, standard treatment of co-existent coronary artery disease is recommended, including lifestyle modification strategies and cardiovascular risk factor management. For patients with coexisting obstructive lesions and symptoms of significant ischemia despite medical intervention, percutaneous or surgical vascularization may be recommended. Previous reports have described the use of coronary artery bypass grafting for the treatment of significant CAD co-existing with ectatic coronary segments.” or page 3 “This is in accordance with recent evidence describing a reduced HDL-C and higher
TG/HDL-C monocyte/HDL-C ratios in CAE and CAD groups as compared to controls." -
page 5 “SM are carried into the vessel wall on lipoproteins and stimulate foam cell formation. They have also been shown to be incorporated in atherosclerotic plaques in addition to the polyunsaturated cholesteryl esters of long-chain fatty acids.”
-
2. The section “Metabolite profiling in CAE” is primarily focusing on the impact of lipids in the pathogenesis of CAE. The authors should either clearly indicate this in the title of this subsection or rename this section to “Lipid profiling in CAE”. Lipidomic profiling is an emerging field due to improved mass spectrometry methods (e.g., ion mobility). The authors should expand this section and describe in more detail at least the two phospholipid classes (SM, PC) that are highlighted. The current version of manuscript does not provide enough information for a broader readership to understand the mechanistic link between lipid remodeling, inflammation and CAE.
3. The manuscript would benefit from a schematic depicting different pathways involved in vascular remodeling during CAE, which are highlighted by the authors.
4. The terms “landmark paper” or “it is believed that” are not clear and subjective. Please remove.
5. Definitions for abbreviations in Figure 1 need to be included in the captions. Furthermore, abbreviations need to be defined throughout the manuscript.
6. Typo: sphyngomyelin (SM) should read sphingomyelin
7. Page 5: “Lower phosphatidylcholine levels have also been implicated in the pathogenesis of CAE. These PC species […]” Please list the PC species or refer to PC class.
Author Response
A major concern are missing references throughout the manuscript. Several important statements by the authors are not clearly supported by underlying citations or identified as comments: f.exp.
- 6: “Overall, standard treatment of co-existent coronary artery disease is recommended, including lifestyle modification strategies and cardiovascular risk factor management. For patients with coexisting obstructive lesions and symptoms of significant ischemia despite medical intervention, percutaneous or surgical vascularization may be recommended. Previous reports have described the use of coronary artery bypass grafting for the treatment of significant CAD co-existing with ectatic coronary segments.” or page 3 “This is in accordance with recent evidence describing a reduced HDL-C and higher
TG/HDL-C monocyte/HDL-C ratios in CAE and CAD groups as compared to controls." - page 5 “SM are carried into the vessel wall on lipoproteins and stimulate foam cell formation. They have also been shown to be incorporated in atherosclerotic plaques in addition to the polyunsaturated cholesteryl esters of long-chain fatty acids.”
We thank the author for these comments. Citing evidence has been updated, including statements without accompany references.
- The section “Metabolite profiling in CAE” is primarily focusing on the impact of lipids in the pathogenesis of CAE. The authors should either clearly indicate this in the title of this subsection or rename this section to “Lipid profiling in CAE”.
Corrected.
Lipidomic profiling is an emerging field due to improved mass spectrometry methods (e.g., ion mobility). The authors should expand this section and describe in more detail at least the two phospholipid classes (SM, PC) that are highlighted. The current version of manuscript does not provide enough information for a broader readership to understand the mechanistic link between lipid remodeling, inflammation and CAE.
- The manuscript would benefit from a schematic depicting different pathways involved in vascular remodeling during CAE, which are highlighted by the authors.
Figure 2 added.
- The terms “landmark paper” or “it is believed that” are not clear and subjective. Please remove.
Corrected.
- Definitions for abbreviations in Figure 1 need to be included in the captions. Furthermore, abbreviations need to be defined throughout the manuscript.
Corrected.
- Typo: sphyngomyelin (SM) should read sphingomyelin
Corrected.
- Page 5: “Lower phosphatidylcholine levels have also been implicated in the pathogenesis of CAE. These PC species […]” Please list the PC species or refer to PC class.
Corrected.
Reviewer 3 Report
The review is interesting. Reporting the characterising aspects of CAE and its diversity respect to aneurysm condition is relevant. The authors stress this comparation, and describe the features of CAE
However, the review needs to be improved by reporting the recent epigenetic and genetic factors involved, as well as the role of gut microbiota, endothelial dysfunction related to vascular ageing, and the consequent vascular remodeling
Author Response
We thank the reviewer for this important suggestion. A new section titled “epigenetics in CAE pathogenesis” has been added to the manuscript. We hope that meets your approval.
Round 2
Reviewer 3 Report
The revised version of your manuscript is sufficiently improved
Author Response
Thank you for your reply and accepting our changes. We much appreciate all contributions and previous comments.
This manuscript is a resubmission of an earlier submission. The following is a list of the peer review reports and author responses from that submission.